# Can We Rely on Self-Assessments of Sense of Coherence? The Effects of Socially Desirable Responding on the Orientation to Life Questionnaire (OLQ) Responses

**Timo Lajunen [1],\* and Esma Gaygısız [2]**

[1] Department of Psychology, Norwegian University of Science and Technology,
   NO-7491 Trondheim, Norway
[2] Department of Economics, Middle East Technical University, 06800 Ankara, Turkey; esma@metu.edu.tr
\* Correspondence: timo.lajunen@ntnu.no

**Abstract:** A large number of studies in health psychology have shown that sense of coherence (SOC) is an essential factor in wellbeing and health. SOC is most commonly measured with the Antonovsky's Orientation to Life Questionnaire (OLQ), which has been so far translated into at least 48 languages. Despite the vast popularity of the OLQ, the relationships between OLQ and socially desirable responding (impression management and self-deception) have not been studied. The purpose of the present study was to investigate the correlations between social desirability and Antonovsky's OLQ. *Method:* The first sample consisted of 423 students who completed the 13-item OLQ and the Eysenck Personality Questionnaire (EPQ), including the Lie scale. Also, the Balanced Inventory for Desirable Responding by Paulhus was administered together with the OLQ to 202 students. *Results:* SOC correlated positively with measures of social desirability among men but not among women. Hence, sex moderated the relationship between socially desirable responding and sense of coherence. *Conclusions*: Socially desirable responding and, especially, self-deception are positively related to high scores in SOC among men but not among women. The OLQ as a measure of sense of coherence can be used among women without worrying about the bias caused by socially desirable responding. When using the OLQ among men, the strong relationship between self-deception and sense of coherence should be taken into account.

**Keywords:** sense of coherence; self-deception; impression management; bias; social desirability; EPQ Lie

## 1. Introduction

Several studies have recently shown that sense of coherence (SOC) (Antonovsky 1987; Antonovsky and Sagy 1986) is an important salutogenic factor related to many aspects of psychological and somatic wellbeing. High SOC scores have been reported to correlate with other measures of generalized perception of self and environment, e.g., locus of control and hardiness (Albino et al. 2016; Flannery et al. 1994; Korotkov and Hannah 1994; Perenc and Radochonski 2016; Skirka 2000; Waite et al. 1999), with psychological adjustment, e.g., state and trait anxiety and depression (Frenz et al. 1993; Gåfvels et al. 2016; Järvholm et al. 2016; Kaiser et al. 1996; Mattisson et al. 2014; Pillay et al. 2015; Schnyder et al. 2000), with perceived work stress (Mackie et al. 2001; Runeson and Norbäck 2005), health, and wellbeing (Cowlishaw et al. 2013; Feldt 1997; Hassmen et

al. 2000; Pallant and Lae 2002), with health behaviors (Speirs et al. 2016; Waite et al. 1999) and risk taking (Lajunen et al. 1998; Lajunen and Summala 1995).

The Orientation to Life Questionnaire (OLQ) (Antonovsky 1987, 1993; Antonovsky and Sagy 1986) was designed to measure three components of SOC: comprehensibility (CO: sense that life events are explicable and predictable), manageability (MA: feeling that one has resources to meet demands in life), and meaningfulness (ME: life demands seen as a challenge rather than a burden). According to Antonovsky's theory (Antonovsky 1987, 1993) and empirical findings (Bernabé et al. 2009; Bonacchi et al. 2012; Feldt et al. 2007; Feldt and Rasku 1998; Gana and Garnier 2001; Moksnes and Haugan 2014) these three components are strongly interrelated and form a common SOC factor. So far, the OLQ has been used in at least 49 different languages and in at least 48 different countries (Eriksson and Mittelmark 2017).

Despite the great interest in SOC among health psychologists, no studies about the role of social desirability bias in the OLQ answers have yet been published. The self-reports of personality, attitudes, and behavior are, however, inaccurate or even biased to some extent, because at least some subjects tend to engage in socially desirable responding (Nederhof 1985; Paulhus 1984; Paulhus 1991; Paulhus and Reid 1991). In addition to deliberate impression management (faking), the self-reports may be biased by self-deception, which can be characterized as a positively biased but subjectively honest self-description (Paulhus 1984; Verkasalo and Lindeman 1994). According to Paulhus (1991), impression management should be controlled when it is conceptually independent of the trait being assessed but still contributes to the self-reported scores of that trait, whereas self-deception should be taken into account, but not necessarily controlled, if it is linked to content variable (Paulhus 1984; Paulhus 1991).

It can be hypothesized that the OLQ is somewhat vulnerable to bias caused by impression management or self-deception, as it relies on the subject's ability to assess his/her feelings and willingness to report them. Earlier studies have suggested that the OLQ answers may be contaminated by emotionality to some degree (Flannery and Flannery 1990; Flensborg-Madsen et al. 2005; Korotkov and Hannah 1994; Korotkov 1993). The effects of social desirability on responses may be even more severe with OLQ than with scales asking about behavior. It is presumably more difficult to get accurate measures of feelings and emotions than of behaviors. This study aims to investigate the role of different types of social desirability in OLQ answers by using two independent samples.

## 2. Method

### 2.1. Participants

The analyses described in the present paper are based on the analysis of two different student samples. The data were collected among psychology undergraduates during lectures. Since the questionnaires were distributed during the class hours, all participants of the lecture filled in a questionnaire. We can say, therefore, that the samples represent well the Finnish undergraduates studying psychology. The students did not receive any reward (money or course credits) for their participation in the study. The respondents did not fill in any identification information (except sex and age) and were assured about anonymity. The sample characteristics are summarized in Table 1.

**Table 1.** Description of the samples and reliability coefficients for the Orientation to Life Questionnaire (OLQ).

|  | Sample 1 | Sample 2 |
| --- | --- | --- |
| Sample characteristics | | |
| N | 423 | 202 |
| Population | students | students |
| Mean age (SD) | 23.8 (16.7) | 24.3 (6.06) |
| Gender distributions | 16.8% male | 31.2% male |
| Alpha and McDonald's omega (in parenthesis) reliability coefficients | | |
| Comprehensiveness (CO) | 0.74 (0.75) | 0.82 (0.83) |

| | | |
|---|---|---|
| Manageability (MA) | 0.62 (0.63) | 0.76 (0.76) |
| Meaningfulness (ME) | 0.73 (0.74) | 0.81 (0.81) |
| Sense of Coherence (SOC) | 0.86 (0.86) | 0.91 (0.91) |
| Eysenck Personality Questionnaire (EPQ) Lie Scale | 0.97 (0.97) | - |
| Self-deception (SD) | - | 0.85 (0.87) |
| Impression Management (IM) | - | 0.81 (0.82) |

## 2.2. Measures

The Finnish translation by Kalimo, Vuori, and Kalimo (Kalimo and Vuori 1990) of the 13-item OLQ was used for both samples. The Finnish version of the Eysenck Personality Questionnaire (EPQ) (Eysenck and Eysenck 1975; Eysenck and Eysenck 1994; Haapasalo 1990) including the Lie scale was administered to sample 1, whereas sample 2 completed the Finnish version of the Balanced Inventory of Desirable Responding (BIDR) by Paulhus (1991). BIDR is composed of impression management (IM) and self-deception scales (SD). The forms were distributed to undergraduate students in classrooms. Participants were asked not to write their names on the questionnaires and were assured that the answers would be treated in strict confidence.

The Cronbach's alpha (Cronbach 1951) and McDonald's omega (McDonald 1999) reliabilities of the OLQ scales (CO, MA, ME, SOC), EPQ-L scale, and BIDR IM and SD scales are listed in Table 1.

## 3. Results

### 3.1. Reliability of the OLQ Scales

The Cronbach alpha and McDonald's omega coefficients were used for assessing the reliability of the scales. The widely referred rule of thumb about reliabilities states that reliabilities over 0.70 are considered "acceptable", reliability coefficients over 0.80 are "good", and reliability coefficients over 0.90 are "excellent" (Nunnally 1978). Table 1 shows good reliability (over 0.80) for the 13-item OLQ scale in both samples. The subscale reliabilities for the CO and ME scale were acceptable in both samples, whereas the reliability coefficients for the MA scale were below 0.70 in sample 1.

### 3.2. Correlations between the EPQ Lie Scale and the OLQ

The EPQ Lie scale was regarded initially merely as a scale to detect faking (Eysenck and Eysenck 1975) but has recently recognized as describing a personality factor denoting some degree of social naivety or conformity. In the present study, the EPQ was first administered together with the OLQ in sample 1.

Correlation coefficients between the EPQ Lie scale and the OLQ scales are presented in Table 2 for men and women, separately. In addition to *p*-values, bias-corrected bootstrap confidence intervals based on 1000 bootstrap samples were calculated to control the possible bias due to the small sample sizes. Table 2 shows that the correlations between the OLQ and the Lie scale were statistically significant, ranging from 0.31 to 0.42 for men but statistically non-significant and close to zero for women. Correlation coefficients corrected for dis-attenuation were also calculated to compute the correlation between EPQ L and OLQ scales without the weakening effect of the measurement error. These dis-attenuated correlations are based on the assumption that all measures are perfectly reliable (alpha coefficient of 1.0). The dis-attenuated Pearson correlation coefficients ranged from 0.40 to 0.50 for men but remained small (0.02–0.03) for women. The dis-attenuated correlations between the EPQ Lie and the OLQ scales indicate that EPQ L is moderately related to sense of coherence among men but not among women.

**Table 2.** Correlations, dis-attenuated correlations, and bias-corrected bootstrap confidence intervals (95%BCI) between the OLQ scales and the questionnaires measuring social desirability among men and women.

| | Men | | | | Women | | | |
|---|---|---|---|---|---|---|---|---|
| | CO | MA | ME | SOC | CO | MA | ME | SOC |
| EPQ-Lie [1] | | | | | | | | |
| Pearson r | 0.42 *** | 0.31 ** | 0.34 ** | 0.41 *** | 0.03 | 0.03 | −0.03 | 0.02 |
| Dis-attenuated r | 0.50 *** | 0.40 *** | 0.40 *** | 0.45 *** | 0.04 | 0.04 | 0.04 | 0.02 |
| 95%BCI upper | 0.24 | 0.08 | 0.17 | 0.23 | −0.05 | −0.03 | −0.09 | −0.06 |
| 95%BCI lower | 0.57 | 0.49 | 0.49 | 0.57 | 0.27 | 0.27 | 0.18 | 0.27 |
| Impression Management [2] | | | | | | | | |
| Pearson r | 0.58 *** | 0.54 *** | 0.52 *** | 0.59 *** | 0.1 | 0.09 | 0.08 | 0.1 |
| Dis-attenuated r | 0.71 *** | 0.69 *** | 0.64 *** | 0.69 *** | 0.12 | 0.11 | 0.1 | 0.12 |
| 95%BCI upper | 0.08 | 0.09 | 0.04 | 0.09 | −0.3 | −0.29 | −0.28 | −0.34 |
| 95%BCI lower | 0.79 | 0.75 | 0.74 | 0.8 | 0.36 | 0.36 | 0.34 | 0.38 |
| Self-deception [2] | | | | | | | | |
| Pearson r | 0.63 *** | 0.60 *** | 0.59 *** | 0.65 *** | 0.06 | 0.04 | 0.11 | 0.07 |
| Dis-attenuated r | 0.75 *** | 0.75 *** | 0.71 *** | 0.74 *** | 0.07 | 0.05 | 0.13 | 0.08 |
| 95%BCI upper | 0.17 | 0.13 | 0.13 | 0.19 | −0.46 | −0.44 | −0.3 | −0.48 |
| 95%BCI lower | 0.8 | 0.78 | 0.78 | 0.82 | 0.35 | 0.34 | 0.38 | 0.39 |

[1] Sample 1; [2] Sample 2; *** $p < 0.001$; ** $p < 0.01$; * $p < 0.05$.

### 3.3. Impression Management, Self-Deception, and the OLQ Scales

The EPQ Lie score refers to a general tendency to embellish one's answers but does not distinguish between deliberate faking and self-deception. These two forms of social desirability, however, reflect different psychological processes and should, therefore, be treated differently. As suggested by Paulhus (1991), IM reflects contamination and should be controlled, whereas SD refers to genuine but positively biased self-description. In the present study, both forms of social desirability were measured together with the OLQ in sample 2.

Table 2 shows that IM correlated positively with the OLQ scales among men (range of correlations: 0.52–0.59) but not among women (range of correlations: 0.08–0.10). The dis-attenuated correlations (range: 0.64–0.71) showed that the OLQ scores were strongly related to IM scores. The IM and OLQ correlations were small and statistically non-significant among women, even when the reliability of the instruments was taken into account.

The correlations between SD and OLQ scales among men were statistically significant and even stronger than between IM and OLQ scales (range: 0.59–0.65). Dis-attenuated correlations ranged from 0.71 to 0.75, which indicates strong relationships between sense of coherence and SD. The correlations between OLQ scales and SD were statistically non-significant and weak among women.

### 3.4. Sex as a Moderator in Social Desirability—The Sense of Coherence Relationship

Table 2 shows that the relationship between social desirability (i.e., EPQ L, BIDR IM, BIDR SD) and OLQ scores was strongly sex-dependent: the relationship was strong among men in both samples and non-existent among women. In order to investigate the moderator effects of sex on social desirability–OLQ relationships, linear regression analyses with a moderator term were performed.

In the first step, EPQ Lie scale scores were standardized, and a sex x EPQ L moderator term was calculated. In the second step, four linear regression models with each OLQ scale score (CO, MA, ME, SOC) as a dependent variable were calculated. In these models, EPQ L score, sex, and moderator (sex x EPQ L) served as independent variables. The results showed a moderator effect for EPQ L–CO relationship (b = 7.41; beta = 0.20; t = 3.67, $p < 0.001$) EPQ L–MA relationship (b = 4.26; beta = 0.15; t = 2.78, $p < 0.01$); EPQ L–ME relationship (b = 6.08; beta = 0.20; t = 3.68, $p < 0.001$), and EPQ L–SOC

relationship (b = 17.74; beta = 0.22; t = 4.01, *p* < 0.001). These moderator effects mean that EPQ L–OLQ relationships are different for men and women, as Table 2 indicates.

In the same way, the BIDR–OLQ relationships were further studied with moderator analysis. In the first step, BIDR IM and SD scale scores were standardized, and two moderator variables were included (sex x BIDR IM and sex x BIDR SD). In the second step, four linear regression models with each OLQ scale score (CO, MA, ME, SOC) as the dependent variable were calculated. In these models, BIDR IM, BIDR SD, sex, and moderators (sex x BIDR IM and BIDR SD) served as independent variables. The results showed a moderator effect for the BIDR SD–CO (b = 3.31; beta = 0.37; t = 2.59, *p* < 0.01) relationship but not for the BIDR IM–CO relationship (b = 0.13; beta = 0.01; t = 0.10, *p* = 0.922). In the BIDR–MA relationship, the moderator effect of BIDR SD was statistically significant (b = 2.71; beta = 0.39; t = 2.63, *p* < 0.01), while no moderator effect of BIDR IM x sex was detected (b = −0.02; beta = −0.00; t = −0.02, *p* = 0.981). A moderator effect for sex was also found in the BIDR SD–ME relationship (b = 2.10; beta = 0.31; t = 2.09, *p* < 0.05), but not for BIDR IM (b = 0.45; beta = 0.07; t = 0.45, *p* = 0.657). The relationship between BIDR scales and total OLQ (SOC) scores followed the same pattern as that observed with the three OLQ subscales. A moderator effect for the BIDR SD–SOC relationship was found (b = 8.12; beta = 0.40; t = 2.75, *p* < 0.01), while no moderator effect on the BIDR IM–SOC relationship was detected (b = 0.55; beta = 0.03; t = 0.19, *p* = 0.853). These results show that the BIDR SD–OLQ relationship is moderated by sex. This moderation was not found for BIDR IM scores, which might be explained by the high correlation between BIDR IM and BIDR SD (r = 0.81, *p* < 0.001).

## 4. Discussion

Several studies have demonstrated SOC to be an important factor contributing to mental and physical wellbeing. Together with the growing evidence of the importance of SOC as a health resource, questions of psychometric properties, validity, and reliability of the OLQ—an instrument developed to measure SOC—have been raised. For example, research findings concerning OLQ factor structure have supported both one-factor and three-factor models (Bernabé et al. 2009; Bonacchi et al. 2012; Ding et al. 2012; Moksnes and Haugan 2014; Sandell et al. 1998).

Reliability analyses of OLQ scales (CO, MA, ME, SOC) in two Finnish samples showed a good level of reliability (0.86 and 0.91) for the 13-item OLQ scale. The only reliability below the 0.70 threshold (0.62) was one of the MA scales in sample 1. Since the alpha reliability coefficient is directly related to the number of items, we can assume that the low reliability is partly explained by the small number of items (four items) in MA scale. While the other subscales (and also MA in sample 2) showed an acceptable level of reliability (over 0.70), the full 13-item OLQ showed better reliability than the subscales. One conclusion from this result is that the total score based on 13-items should be used instead of less reliable subscales. While this conclusion is certainly correct in terms of Antonovsky's (1987) view of SOC as a holistic concept, earlier studies show that SOC based on CO, MA, and ME shows high construct validity and reliability (Lajunen 2018). Considering earlier studies and the results of the current study, we can conclude that the choice between one OLQ total score and three OLQ scale scores should be based on the aims of the study in which OLQ is used. If the aim is to measure CO, MA, and ME scores, it is advisable to use the 29-item OLQ instead of the short 13-item scale.

Despite the vast popularity of SOC and a vast interest in the psychometric properties of OLQ, the role of social desirability in SOC has remained unexamined. In the present study, the influence of social desirability on SOC scores was addressed by examining correlations between the OLQ and the EPQ Lie scale (Eysenck and Eysenck 1975) and BIDR (Paulhus 1991). The results showed that almost all OLQ scale scores were somewhat biased by social desirability among male respondents. The disattenuated correlations between EPQ Lie scale and OLQ scales ranged from 0.40 to 0.50, which shows that the OLQ scores are moderately influenced by EPQ Lie scores. Whether this can be considered as a "contamination" by lying depends on how EPQ Lie scores are interpreted.

According to Paulhus (1991), IM score refers to the deliberate tendency to give favorable self-descriptions to others and comes, therefore, close to lying and falsification. The present study showed that male participants scoring high in IM also scored high in OLQ (r = 0.52–0.59). When the correlation

coefficients were corrected for the effect of the measurement error, they ranged from 0.64 to 0.71, which indicates a severe effect of impression management on the sense of coherence scores (or vice versa). This result emphasizes the importance of paying attention to factors which might increase the probability of impression management and, thus, decrease the reliability of the OLQ. In the present study, the participants were assured that their responses would be treated in the strictest confidence. In addition, the participants answered the questionnaires anonymously. These procedures naturally minimize the bias caused by IM. This kind of total anonymity based on testing a large group simultaneously might not be easily achieved, say, in clinical settings where OLQ is administered to patients. Therefore, the possibility of distortion in OLQ scores should be kept in mind and corrected statistically if necessary.

In contrast to IM, self-deception as an unintentional over-positive view of oneself might help individuals to cope with the demands of everyday life. SD has been found to be intrinsically linked to positive personality constructs such as psychological adjustment (Paulhus and Reid 1991; Taylor and Brown 1988), high self-esteem (Paulhus and Reid 1991; Robins et al. 2001), and lack of neuroticism (Davies et al. 1998; Otter and Egan 2007). In the present study, similar relationships between SD and SOC were found. Among men, the correlations between OLQ scale scores and self-deception were even higher than the ones between impression management and sense of coherence. It is possible that a positive relationship between SOC and mental and physical wellbeing can partly be explained by a link between positive attitude to oneself and health. Hence, having a high SOC might even require a slightly positively biased attitude to oneself.

Correlations between the sense of coherence and all measures of socially desirable responding, including EPQ Lie scores and BIDR impression management and self-deception scores, were high among men but not among women. Moderator analyses showed that sex moderated all relationships between EPQ Lie scores and BIDR IM and SD scores: the effect of socially desirable responding was found for men but not for women. Interestingly, when the effects of impression management and self-deception scores on the sense of coherence were analyzed by using regression analysis with a moderator variable (sex x BIDR scale), the moderator effects were found only for self-deception. This finding, together with high correlations, on the one hand, between self-deception and sense of coherence and, on the other hand, between self-deception and impression management indicate that the effect of social desirability on the sense of coherence is predominantly caused by self-deception, which also explains a large proportion of variance in impression management. Hence, the sense of coherence scores was not "contaminated" by lying or deliberate impression management but instead reflected an overly positive view of oneself which can be labeled as self-deception. This is understandable because the participants would not have benefitted from embellishing their answers.

It is interesting that the socially desirable responding (self-deception) was related to sense of coherence among men but not among women. This result might indicate that the origins of the sense of coherence—i.e., seeing one's life manageable, meaningful, and understandable—are different for Finnish men with respect to for Finnish women. Another possibility is that men and women have different response styles, which is reflected in both measures of social desirability and sense of coherence. For example, especially young men might be more likely than women to choose extreme values in measures of social desirably and sense of coherence. Men endorsing a more traditional masculine gender role (e.g., assertiveness, achievement, and success) might score higher in both socially desirable responding and sense of coherence in order not to reflect their weaknesses or uncertainty. Earlier research has consistently demonstrated that men score higher on egoistic response tendencies and that women score higher on moralistic response tendencies (Lalwani et al. 2006). Thus, men and women are susceptible to different types of socially desirable responding (Steenkamp et al. 2010), which can also be reflected in OLQ responses. The different types of gender-related socially desirable responding may also be reflected in the inconclusive findings of sex differences in SOC scores reported in various studies. While one-third of the studies about SOC in adolescence have not found any differences between sexes, about two-thirds of the studies have reported a higher level of SOC among boys (Rivera et al. 2013). Since the present study did not include any gender role measures and was conducted only within one culture/country, it is impossible to

explain why the male sample differed from the female one. It can only be said that the OLQ as a measure of sense of coherence might work differently among men and women, at least in Finland.

Some limitations of the study should be taken into account. While the gender distribution of the sample reflects the gender distribution in the population (psychology students), a small number of men in the sample prevents more detailed analyses about gender differences in socially desirable responding, response styles, and OLQ responses. Large and a more heterogeneous samples would allow the investigation of socially desirable responding and sense of coherence in different age groups. Moreover, the present study was a cross-sectional study in which the sense of coherence scores was correlated with indicators of socially desirable responding. This kind of mono-cultural study based on an ecological design (correlations) can reveal interesting relationships but is vulnerable to methodological biases. First of all, this study was based on student samples collected in one country (Finland). Sense of coherence is a highly culturally dependent concept, as any measure of meaning in life. Finnish respondents might emphasize different aspect as meaningful, comprehensible, or manageable than, say, people from more collectivist cultures. In the future, cross-cultural studies of the sense of coherence and its correlates are needed. Secondly, this study was based on surveys, which might increase the method bias in responses. While a multitrait-multimethod matrix (MMM) approach would be a much stronger design for detecting the possible effects of social desirability on the sense of coherence, it is challenging to design studies in which the sense of coherence and social desirability could be measured with various methods rather than with surveys only. In future studies, different methods for studying the effects of socially desirable responding on the sense of coherence should be investigated.

**Author Contributions:** Conceptualization, T.L.; Data curation, T.L.; Formal analysis, T.L. and E.G.; Investigation, T.L. and E.G.; Methodology, T.L. and E.G.; Writing – original draft, T.L.; Writing – review & editing, E.G.

**Funding:** This research received no external funding.

**Conflicts of Interest:** The authors declare no conflict of interest.

## Abbreviations

| | |
|---|---|
| SOC | Sense of Coherence |
| OLQ | Orientation to Life Questionnaire |
| CO | Comprehensibility |
| MA | Manageability |
| ME | Meaningfulness |
| EPQ | Eysenck Personality Questionnaire |
| BIDR | Balanced Inventory of Desirable Responding |
| IM | Impression Management |
| SD | Self-Deception scales |

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
