# Peer review of "Can We Rely on Self-Assessments of Sense of Coherence? The Effects of Socially Desirable Responding on the Orientation to Life Questionnaire (OLQ) Responses"

_socsci, doi:10.3390/socsci8100278_

Round 1

Reviewer 1 Report

SOC is an important indicator of physical and mental health and a coping resource. The association between the Orientation to Life Questionnaire by Antonovsky and the lie Scale by Eysenck and the impression management and self-deception measurements by Paulhus respectively have not been studied yet.

The paper is well structured, the statistics and results are soundly presented. The interesting results show gender differences indicating that men prefer a response style of impression management and self-deception whereas women tend to endorse the SOC-Questionnaire and the Social Desirability Questionnaire without obvious response bias.

In the discussion the authors don’t discuss gender differences in response styles – and this is missing. They argue not to have used a gender role measurement – but all questionnaires are related by the kind of measurement and might be associated by response bias. There are a lot of papers which show that men use to describe themselves more positive than women in health issues, in self-esteem etc. – without control of social desirability and behavior measures, results in questionnaires cannot be counted as behavior or personality trait as such, but influenced by response bias.

Author Response

Thank you for your constructive comments, which helped us to improve the manuscript.

Reviewer: “In the discussion the authors don’t discuss gender differences in response styles – and this is missing. They argue not to have used a gender role measurement – but all questionnaires are related by the kind of measurement and might be associated by response bias. There are a lot of papers which show that men use to describe themselves more positive than women in health issues, in self-esteem etc. – without control of social desirability and behavior measures, results in questionnaires cannot be counted as behavior or personality trait as such, but influenced by response bias.”

Our response:

Thank you for this interesting and important comment. Because of the importance of this issue, we decided to add a paragraph with some new references to the Discussion section (page 6, lines 233-253).

Reviewer 2 Report

I find this article interesting with clear findings. It is of interest and could help with the interpretation of self-rating questionnaires. The article is well referenced ( some authors have though suggested that only studies that use the same version of the OLQ should be compared).

I have a few comments.

In the introduction it is stated that the aim of the study is to investigate the role of the different types of social desirability in OLQ answers using four independent samples. 

Table 1 shows 2 different samples.

The authors might comment about the gender distribution since the majority of the participants were females. How about the attrition and how representative were the male and female  respondents ? Can the samples and the selection mechanism be described in more detail?

The results are evidently described, stating that the males have a strong relationship between self-deception and the way of answering OLQ. I agree with the authors that it is impossible to explain why the the male sample differed from females, and that this should be further investigated.

The respondents were also quite young, is it possible that young persons are more prone to answer in a socially desirable way? Is this investigated?

The discussion section is satisfactorily, underlining that despite the vast interest in OLQ the role of social desirability has remained unexamined. The comments about mono-cultural studies and the vulnerability to methodological biases are valuable.

Author Response

Thank you for your comments, which were very helpful for revising the manuscript.

Reviewer: “In the introduction it is stated that the aim of the study is to investigate the role of the different types of social desirability in OLQ answers using four independent samples. Table 1 shows 2 different samples.”

Response: We have corrected this typo to “two independent samples”.

Reviewer: “The authors might comment about the gender distribution since the majority of the participants were females. How about the attrition and how representative were the male and female respondents ? Can the samples and the selection mechanism be described in more detail?”

Response: Thank you for this comment. We added description of the sampling (and representativeness) to the “method” section and also added a “shortcomings” section to Discussion.

Reviewer: “The respondents were also quite young, is it possible that young persons are more prone to answer in a socially desirable way? Is this investigated?”

Thank you for pointing this out. We added a paragraph to Discussion, in which we speculate about different types of gender-related socially desirable responding and also mention age as one factor (page 6, lines 233-253). We also added a suggestion for future research to investigate age effects to the “limitations“ section in Discussion (page 7, lines 257-258).